# LEARNING TO GIVE CHECKABLE ANSWERS WITH PROVER-VERIFIER GAMES

## ABSTRACT

Our ability to know when to trust the decisions made by machine learning systems has not kept up with the staggering improvements in their performance, limiting their applicability in high-stakes domains. We introduce Prover-Verifier Games (PVGs), a game-theoretic framework to encourage learning agents to solve decision problems in a verifiable manner. The PVG consists of two learners with competing objectives: a trusted verifier network tries to choose the correct answer, and a more powerful but untrusted prover network attempts to persuade the verifier of a particular answer, regardless of its correctness. The goal is for a reliable justification protocol to emerge from this game. We analyze variants of the framework, including simultaneous and sequential games, and narrow the space down to a subset of games which provably have the desired equilibria. To complement our novel learning problem, we propose rigorous methodology for evaluating the robustness of the learned protocol. We develop instantiations of the PVG for two algorithmic tasks, and provide empirical evidence that the verifier can in fact learns a robust decision rule that is able to receive useful and reliable information from an untrusted prover. Importantly, the protocol still works even when the verifier is frozen and the prover's messages are directly optimized to convince the verifier.

## 1 INTRODUCTION

The astonishing performance of today's dominant learning paradigm – optimizing powerful differentiable function approximators to minimize suitable loss functions – often comes at the cost of poor robustness and reliability. It is common for powerful deep learning systems to be vulnerable to adversarial attacks (Goodfellow et al., 2015), to display erratic behaviour on out-of-distribution data and be very confident of wrong predictions (Che et al., 2021). How can we train our learning algorithms to produce outputs that can be checked by humans or by learning algorithms that are cheaper or better understood?

Over the past decades, the field of computational complexity has greatly expanded our notion of proof to include formalisms such as interactive, zero-knowledge, and probabilistically checkable proofs (Goldreich, 2008). Most of these notions can be thought of in terms of a game between a powerful but untrusted prover and a computationally limited but trusted verifier. Taking inspiration from this, we propose the Prover-Verifier Game (PVG), where two learning agents play the role of prover and verifier, and are allowed to converse. The verifier aims to determine the correct answer to a decision problem, and the prover aims to convince the verifier of a particular answer (regardless of its correctness). Since the prover is untrustworthy, the verifier will only find its messages useful to the extent that it can independently verify the information. If all goes well, then the game dynamics will lead to a proof protocol which, if not mathematically sound, is at least sufficient for the whole system to achieve more reliable predictions than the verifier can achieve unaided.

We analyze the desirability of the game equilibria implied by several variants of the Prover-Verifier Game concept and narrow the space down to a subset of games that theoretically have the desired equilibria. Picking the right game equilibrium concept turns out to be essential: we prove that formulating the PVG as a sequential game in which the prover agent plays first leads to dysfunctional solutions. On the other hand, the simultaneous (connected to Nash equilibria) and verifier-first sequential formulations (connected to verifier-leading Stackelberg equilibria) have desirable equilibria. We formally show, on a illustrative problem, that gradient based differentiable game optimizers can find desirable proof-verification protocols.

Do reliable justification protocols emerge in practice if we let artificial agents play the Prover-Verifier Game? To complement our novel game formulation, we develop a rigorous evaluation methodology whereby the verifier is frozen and the prover (or the message) is optimized to convince the verifier. We then run simulations on two algorithmic tasks using a practical instantiation of the PVG concept. As predicted by our theory, PVG-trained verifiers learn to receive useful and reliable information from untrusted provers by following sensible verification protocols, whereas those trained alongside fully collaborative provers are easily deceived.

## 2 BACKGROUND

### 2.1 INTERACTIVE PROOF SYSTEMS (IPS)

Interactive proof systems generalize the notion of a mathematical proof to a dialogue between two agents - a prover and a verifier - to solve a decision problem (Arora & Barak, 2009; Thaler, 2019). The verifier agent, who is trustworthy but computationally constrained, is tasked with producing a correct answer to the decision problem. It can exchange messages with a computationally unbounded, yet potentially adversarial prover agent. The communication protocol between the prover and verifier constitutes an *interactive proof system* if and only if it is sound and complete:

**Definition 1** (Completeness and Soundness). *The verifier of an interactive proof system is complete iff there exists a prover that can always convince the verifier that the answer is "yes", if the correct answer is "yes". It is sound iff there doesn't exist a prover that can trick the verifier into answering "yes" if the correct answer is "no".*

### 2.2 DIFFERENTIABLE GAME OPTIMIZATION

A two-player differentiable game consists of two agents whose strategies are parametrized by $\mathbf{w} = (\mathbf{w}_1, \mathbf{w}_2) \in \mathbb{R}^d$ that take turns to minimize their differentiable loss functions $(\mathcal{L}_1, \mathcal{L}_2) : \mathbb{R}^d \to \mathbb{R}$. An *equilibrium concept* determines which strategies will be adopted by the players. A *Nash equilibrium* is achieved when no player can unilaterally improve its objective function.

**Definition 2** (Nash Equilibrium (Von Neumann & Morgenstern, 2007)). *The strategies parametrized by $(\mathbf{w}_1^*, \mathbf{w}_2^*)$ constitute a Nash equilibrium[1] of the two-player sequential differentiable game with loss functions $(\mathcal{L}_1, \mathcal{L}_2) : \mathbb{R}^d \to \mathbb{R}$ if they minimize their loss functions keeping the other player's parameters fixed:*

$$\mathbf{w}_1^* = \arg\min_{\mathbf{w}_1} \mathcal{L}_1(\mathbf{w}_1, \mathbf{w}_2^*), \quad \mathbf{w}_2^* = \arg\min_{\mathbf{w}_2} \mathcal{L}_2(\mathbf{w}_1^*, \mathbf{w}_2) \tag{1}$$

The notion of equilibrium considered by Generative Adversarial Networks (Goodfellow et al., 2014) is an example of a Nash Equilibrium. A *Stackelberg equilibrium* differs from the Nash equilibrium in that one of the players is deemed the "leader" and the other the "follower" (Wang* et al., 2020). It is assumed that the follower always picks the optimal strategy for a given leader strategy. In response, the leader modifies its strategy by factoring in how the follower agent will respond to the modification.

**Definition 3** (Stackelberg Equilibrium (Fiez et al., 2020)). *Let $\mathbf{w}_1$ parametrize the strategy of the "leader agent" and $\mathbf{w}_2$ parametrize that of the "follower agent". The loss functions of the agents are $\mathcal{L}_1$ and $\mathcal{L}_2$ respectively. The strategies parametrized by $(\mathbf{w}_1, \mathbf{w}_2)$ constitute a Stackelberg equilibrium[1] if and only if (1) the follower's strategy is optimal given the leader's strategy, and (2) the leader's strategy is optimal taking into consideration how the follower will respond to modifications.*

$$\mathbf{w}_1^* = \arg\min_{\mathbf{w}_1} \mathcal{L}_1(\mathbf{w}_1, \mathbf{w}_2^*(\mathbf{w}_1)), \quad \mathbf{w}_2^*(\mathbf{w}_1) = \arg\min_{\mathbf{w}_2} \mathcal{L}_2(\mathbf{w}_1^*, \mathbf{w}_2) \tag{2}$$

## 3 PROVER-VERIFIER GAME

The Prover-Verifier Game aims to learn decision rules with a reliable internal verification step. We describe what we mean for a verification protocol to be reliable, then outline the game formulation.

### 3.1 DESIDERATA AND PROVER-VERIFIER INCENTIVES

**Desiderata:** Designing the right prover-verifier game requires precisely defining what a desirable outcome/equilibrium of the game looks like. With the "completeness" and "soundness" definitions in mind (Section 2.1), we list the properties we seek in a desirable verifier protocol:

---

[1]We assume uniqueness of equilibria for simplicity.

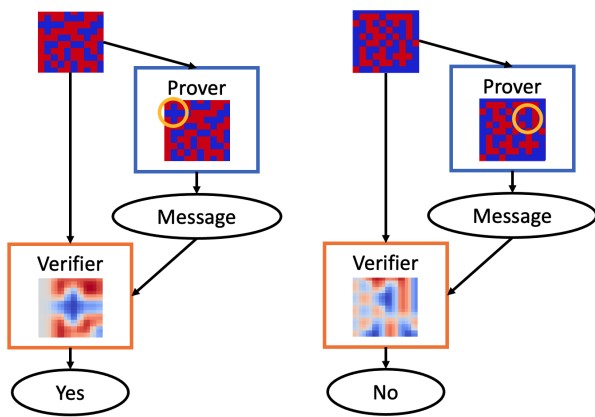

(a) Pattern to detect.     (b) When the pattern exists.     (c) When the pattern doesn't exist.

Figure 1: Depiction of a proof-verification protocol learned via Prover-Verifier Training. When trained on the task of detecting whether there's a blue plus pattern (1a) in the input image, Prover-Verifier Training discovers a coordinate-based verification protocol. If there's a plus in the image (1b), the prover sends its coordinate, and the verifier accepts this certificate. If there's not a plus in the image (1c), then the prover sends the coordinates of a maximally convincing image patch, but the verifier rejects this certificate.

- **Possibility of Perfect Recall:** There should exist a prover that can help the verifier achieve perfect recall — the ratio of true positive predictions to all positive examples.
- **Guarantee of Perfect Precision:** There shouldn't exist any prover which can trick the verifier into achieving non-perfect precision — the ratio of true positive predictions to all positive predictions.

"Possibility of perfect recall" is connected to completeness, and implies that with the right proofs, the verifier can achieve zero false negative rate. "Guarantee of perfect precision" is related to soundness and implies that the verifier always has zero false positive rate regardless of which proof is used.

**Picking the right prover-verifier incentives:** We propose to set up the prover-verifier game with the following incentive structure: the verifier is incentivized to always give the correct answer to the given decision problem, and the prover is incentivized to get the verifier to answer "yes" to the decision problem, regardless of the correct answer. This structure encourages collaborative dynamics when the correct answer to the decision problem is "yes" (which is linked to recall) and encourages adversarial dynamics if the correct answer is "no" (which is linked to precision). As we will prove formally in Section 3.3, this incentive structure, when embedded in the right game setup with the right loss functions, can lead to a proof-verification protocol that meets the desiderata.

## 3.2 Different Prover-Verifier Game Formulations

**Notation** Let $x \sim p_X(x)$, $y \sim p_Y(y)$ and $y' \sim p_{Y'}(y')$ be the input, verifier label and prover label random variables where we have $x \in \mathbb{R}^{n_x}$, $y \in \{0, 1, ..., K\}$ and $y' \in \{0, 1, ..., K\}$. Let their joint distribution be $p_D$.

The prover agent $\mathcal{P}_{\mathbf{w}_p}$, parametrized by $\mathbf{w}_p \in \mathbb{R}^{n_p}$, outputs the conditional probability distribution $p_p(z|x)$ over real valued "message" vectors $z \in \mathbb{R}^{n_z}$ conditioned on inputs. The verifier agent $\mathcal{V}_{\mathbf{w}_v}$, parametrized by $\mathbf{w}_v \in \mathbb{R}^{n_v}$, represents the conditional probability distribution $p_v(\hat{y}|x, z)$ over the predicted labels $\hat{y} \in \{0, 1, ..., K\}$, conditioned on the inputs $x$ and messages $z$.

**Loss Functions:** We pick the following loss functions for the verifier and prover:

$$\mathcal{L}_v = \mathbb{E}_{(x,y,y')\sim p_D,\ z\sim p_p(z|x)}[-\log p_v(y|x, z)] \tag{3}$$

$$\mathcal{L}_p = \mathbb{E}_{(x,y,y')\sim p_D,\ z\sim p_p(z|x)}[-\log p_v(y'|x, z)] \tag{4}$$

Since we wish for the verifier agent to try and solve the problem to the best of its ability, we set $p_Y$ to be the correct label distribution. Since we wish to incentivize the prover to try and defend a

particular answer regardless of its correctness, we set $p_{Y'}(y') = 1$ where $y' \in \{0, 1, \ldots, K\}$ is the output that the prover is defending. In a decision problem, $y'$ will be one of 0 or 1. One can consider variations on these loss functions by replacing the negative log with other functions that are convex and monotonically increasing. We prefer the aforementioned loss functions when the verifier agent represents a softmax policy over its outputs. This guarantees that the gradients vanish iff the agents fulfill their incentives (see Appendix A).

**Suitable Equilibrium concepts:** There are three obvious ways to set up the order in which the agents play (i.e. pick their strategies parametrized by $\mathbf{w}_p$ and $\mathbf{w}_v$) the Prover-Verifier Game: 1) The prover and verifier play simultaneously, 2) The prover plays first, 3) The verifier plays first. The simultaneous setup leads to picking Nash equilibrium as the equilibrium concept. The latter two lead to picking Stackelberg equilibrium, where prover or verifier is the leader and can therefore reason about how the opponent will respond to a given strategy. Interestingly, as we show in Section 3.3, not all of these formulations lead to equilibria that satisfy our desiderata.

**When Instances are Revealed:** We can also arrive at different game formulation depending on whether the problem instances (i.e., the input and prover-verifier labels) are revealed to the agents before or after they pick their strategies (i.e. weights). This leads to eight different game formulations: two from the simultaneous setup (before or after the simultaneous moves) and six from the sequential setup (before, in-between and after the moves, for each of the two sequential formulations).

**How the Prover and Verifier Interact:** The design space of how the prover and verifier interact with each other is large. We only consider the single-step, no feedback formulation throughout the paper and leave the analysis of other formulations as future work. Note that the single-step, no feedback formulation is related to NP proof systems and include sophisticated proof-verification protocols. We can also derive numerous PVG formulations by modifying the communication channel between the prover and verifier. Various decisions regarding the channel include: 1) If the channel is stochastic or deterministic (and what kind of noise there is) 2) Whether prover's messages are real valued vectors, or discrete tokens 3) Whether the prover and verifier use an already constructed communication protocol, such as natural language.

### 3.3 PVG FORMULATIONS WITH DESIRABLE EQUILIBRIA

We list in Table 1 all possible game instantiations obtained by varying the 1) player order and 2) when the problem instance is revealed. Our goal is to find which – if any – of these formulations have desirable equilibria. We consider the following abstract decision problem formulation to compare and contrast different PVG formulations:

**Definition 4** (Abstract Decision Problem). *Let $\mathcal{X}$, $\mathcal{M}$, $\mathcal{Y} = \{0, 1\}$ and $\tilde{\mathcal{Y}} = \{0, 1\}$ be the (discrete) sets of all inputs, prover messages, and labels, respectively. Let $\mathbb{P}_X$ represent the input distribution. Assume the label is a deterministic function of the input, $y = f(x)$, and the prover is always arguing for the label $y' = 1$. The prover agent can choose a policy $p \in \mathcal{P}$, where $\mathcal{P}$ is a set of mappings $\mathcal{X} \to \mathcal{M}$. Let $\mathcal{V}$ denote a set of base verifiers, which are mappings $\mathcal{X} \times \mathcal{M} \to \{0, 1\}$. The verifier chooses a policy $v$ which computes a convex combination of elements of $\mathcal{V}$. (The verifier's output can be interpreted as the probability it assigns to the label being 1). The verifier and prover try to minimize the loss functions defined in Equations 3 and 4 respectively. We assume that both $\mathcal{P}$ and $\mathcal{V}$ include every constant mapping (i.e. ones which don't depend on the input).*

*Assume that there exists a deterministic verifier policy $v^* \in \mathcal{V}$ such that:*

$$\exists p \in C_p \colon \ \forall x \in \mathcal{X} \colon \ v^*(x, p(x)) = f(x) \ \ \textit{(completeness)} \tag{5}$$

$$\forall p \in C_p \colon \ \forall x \in \mathcal{X} \colon \ v^*(x, p(x)) = 1 \implies f(x) = 1 \ \ \textit{(soundness)} \tag{6}$$

For instance, to represent a decision problem from complexity theory, we could take $\mathcal{X}$ and $\mathcal{M}$ to be sets of binary strings, $\mathcal{P}$ to be the set of all mappings $\mathcal{X} \to \mathcal{M}$ (the prover is typically computationally unrestricted), and $\mathcal{V}$ the set of mappings implementable by boolean circuits with some maximum size. However, we keep the formalism general because our main results don't depend on the details of boolean circuits or complexity theoretic assumptions.

The conclusions of our theoretical analysis are displayed in Table 1. Proper proof-verification protocols constitute equilibria of only three of the eight game formulations: the simultaneous setup where the problem instance is presented after the agents' moves, and the two verifier-first sequential setups where the problem instances are revealed after the verifier's move. We justify these claims through the following theorems. The proofs are presented in Appendix B.

Table 1: The summary of our theoretical results on the desirability of equilibria of different PVG formulations. The "ordering" column illustrates in which order the prover and verifier pick their strategies (weights) and the problem instance (the input and labels) are revealed. The "necessity" and "sufficiency" columns show whether having converged to an equilibrium is necessary and/or sufficient for having discovered and complete and sound proof-verification protocol. The "failure mode" column describe in what way some of the setups lead to undesirable equilibria. The "convergence" column specifies if gradient descent converges to the eqlibrium on the Binary Erasure Channel Task. (\*)We've only shown local convergence for the verifier-leading equilibria.

| Ordering | Necessity | Sufficiency | Failure Mode | Convergence |
|---|---|---|---|---|
| {Prover, Verifier}, Instance | ✓ | ✗ | coordination problem | ✓ |
| Instance, {Prover, Verifier} | ✗ | ✗ | trivial verifier | ✗ |
| Verifier, Prover, Instance | ✓ | ✓ | none | ✓* |
| Verifier, Instance, Prover | ✓ | ✓ | none | ✓* |
| Instance, Verifier, Prover | ✗ | ✗ | trivial verifier | ✗ |
| Prover, Verifier, Instance | ✗ | ✗ | flood-the-zone | ✗ |
| Prover, Instance, Verifier | ✗ | ✗ | trivial verifier | ✗ |
| Instance, Prover, Verifier | ✗ | ✗ | trivial verifier | ✗ |

**Proposition 1.** *Any PVG formulation in which the verifier is given the problem instance before it selects its strategy has bad equilibria — that is, having converged to an equilibrium is neither necessary nor sufficient for having found a complete and sound proof verification protocol.*

If the instance $x$ is known when the verifier chooses its policy, then the verifier can simply choose the constant policy which returns $f(x)$. We dub this the "trivial verifier" failure mode.

**Theorem 1.** *Consider all prover-leading sequential PVG formulations. Having reached a (prover leading Stackelberg) equilibrium is neither necessary nor sufficient for having found a complete and sound proof-verification protocol.*

The intuition behind this result is as follows: no matter what strategy the prover picks as the leader, the verifier can always find a strategy whose outputs match the label marginals (i.e. at least $50\%$ accuracy if the labels are balanced). Due to the algebraic form of the prover loss function, the best the prover can do is to avoid giving the verifier any information about the information whatsoever about the label. This defective incentive prevents the learning of proper proof-verification protocols. We dub this the "flood-the-zone" failure mode, in reference to the media disinformation strategies that pollute the information ecosystem to prevent people from knowing what's true Illing (2020).

**Theorem 2.** *Consider all verifier-leading sequential PVG formulations in which the problem instance is revealed after the verifier picks its strategy. Having reached a (verifier leading Stackelberg) equilibrium is both necessary and sufficient for having found a desirable proof-verification protocol.*

Necessary is easy to argue: given a complete and sound protocol, neither player has an incentive to deviate. Sufficiency can be shown using a "proof by contrapositive" argument: If the protocol represented by prover and verifier is not complete and sound, then they cannot possibly represent a verifier-leading Stackelberg equilibrium, because the verifier can do strictly better by switching to the complete and sound verification system (as the prover will react by producing correct proofs).

**Theorem 3.** *Consider the simultaneous PVG formulation where the problem instance is revealed after the agents pick their strategies. Having converged to a (Nash) equilibrium is necessary for having found a complete and sound proof-verification protocol.*

If the prover and verifier already represent a complete and sound proof-verification protocol, the verifier's loss function is already minimized, hence the verifier won't change its strategy. Because the prover can already convince the verifier when it's acting collaboratively (due to completeness) and cannot fool it when it's acting adversarially (due to soundness), it also won't have the incentive to change its strategy. Therefore, the prover-verifier strategies constitute Nash equilibria. This also implies that all Stackelberg equilibria of the proof-verification game are also Nash equilibria:

**Corollary 1.** *All verifier-leading Stackelberg equilibria are Nash equilibria of PVG.*

There might exist Nash equilibria that don't correspond to complete and sound protocols. For instance, the case in which the verifier completely ignores the prover constitutes a Nash equilibrium. We call this the "coordination problem" failure mode: no player can adopt a protocol unilaterally.

### 3.4 Finding Proof-Verification Strategies Using Gradient-Based Optimizers

Section 3.3 establishes that there exist PVG formulations whose equilibria, if found, will constitute complete and sound proof-verification protocols. Is there hope in finding these equilibria using gradient based differentiable game optimizers? We thereon use the phrase "Prover-Verifier Training" to refer to the general method of using gradient-based optimizers to solve for PVG equilibria.

To address the question above, we turn to a simpler problem we call "Binary Erasure Channel" (BEC) which admits detailed analysis. The BEC problem is defined as follows:

**Definition 5.** *We have a prover-verifier system. The prover takes in input $x \in \{0, 1\}$ and attempts to convince the verifier to output 1 all the time whatever the input is. The verifier aims at predicting the input correctly only based on the message from the prover. The prover defines a stochastic function $f : \{0, 1\} \to \{0, 1, ..., K\} \triangleq \mathcal{M}$ and is parameterized by probability vectors $p^0 = [p_0^0, ..., p_K^0]$ and $p^1 = [p_0^1, ..., p_K^1]$ while the verifier defines $g : \mathcal{M} \to \{0, 1\}$ and is parameterized by $q \in \mathbb{R}^{K+1}$ with $q_i$ representing the probability of predicting 1 when receives token $i \in \mathcal{M}$.*

We impose some further restrictions: the prover cannot send the message 1 when $x = 0$ and cannot send 0 when $x = 1$. With this restriction, the communication channel between the prover and verifier resemble the *binary erasure channel* Cover (1999) with the tokens $\{2, ..., K\}$ representing multiple erasure tokens. This setup is a simplification of the more general assumption that there exists a complete and sound verification protocol that is realizable by the prover and verifier. In this setting (given loss functions in Eq. equation 3 and equation 4), the joint strategy of prover sending the right proof with $p_1^1 = 1$ and verifier proceeding with $q_1 = 1$ but $q_i = 0$ for all $i \neq 1$ is a Nash equilibrium.

Moreover, we add entropy regularization (i.e., label smoothing) to the verifier loss to avoid degenerate cases in training. We can show that running standard gradient descent converges to the ideal joint strategy under the simultaneous game formulation. In more detail, we have the following theorem.

**Theorem 4.** *Starting from any initialization with $p_i^0 > 0, p_i^1 > 0$ for all $i \in \mathcal{M}$, running gradient descent on both prover and verifier with reasonable learning rates, the prover would converge to only sending message 1 when $x = 1$ while the verifier would predict 1 only if it receives message 1 from the prover. To put it differently, $p_1^1 = 1$ when converge while $q_1 = 1 - \epsilon$ and $q_0 = q_2 = ... = q_K = \epsilon$ with $\epsilon$ a small constant depending on the strength of entropy regularization.*

Formulating the PVG as a sequential game in which the prover agent plays first leads to undesirable solutions. We summarize the the result below.

**Theorem 5.** *Starting from any initialization with $p_i^0 > 0, p_i^1 > 0$ for all $i \in \mathcal{M}$, running gradient descent on both prover and verifier with any learning rates, prover would converge to only sending message 2 to $K$ no matter what the input is.*

## 4 Related Work

**Computational Complexity:** Much of computational complexity theory is centered around various notions of proofs, from the basic definition of NP to more modern notions of proof such as interactive, zero-knowledge, and probabilistically checkable proofs (Goldreich, 2008). These various notions of proof can be understood in terms of a game between a prover and verifier somewhat analogous to ours. Our game is most directly analogous to the vanilla notion of proof used to define NP. But given the right architecture, our game would allow the prover's message to encode the weights of a network that answers the verifier's questions, thereby more closely resembling an interactive proof.

**Interpretability:** Selective rationalization is the task of highlighting input features that are useful for classification. A number of game theoretic selective rationalization methods have been proposed (Lei et al., 2016; Yu et al., 2019; Chang et al., 2019). With text applications in mind, Lei et al. (2016) propose a two player, collaborative game where a generator network selects relevant features from the input text, and a predictor network does classification using those features. Chang et al. (2019) propose a six player game where the 4 generators select factual or counterfactual rationales, and the 2 discriminators aim to distinguish factual and counterfactual rationales. In practice, only 3 players are instantiated (two generators that only take negative or positive examples and a discriminator). Game theoretic approaches have also been used for interpretable fact-checking on knowledge graphs. Hildebrandt et al. (2020) propose solving the triplet classification task (deciding whether a subject-predicate-object triple is contained in a knowledge graph) using debate dynamics. Two reinforcement

learning agents learn to generate arguments (paths in a knowledge graph) that either support or go against a fact (a triple). These are then fed to a classifier trained to ascertain the truth value of the fact.

**Verifying correctness and functional properties of models:** There's a growing body of literature aimed at verifying both the correctness and/or functional properties of learned models. Goldwasser et al. (2021) investigate interactive proof systems to verify correctness. They show cases where PAC[2] verification of hypotheses are significantly more sample efficient, even when using a single-round protocol. Works that aim to verify functional properties of models (such as robustness to adversarial perturbations) can be categorized under three groups (Kohli et al., 2019): those that test consistency with specifications (Ruderman et al., 2018; Uesato et al., 2018), those that train specification consistent models (Raghunathan et al., 2018; Wong et al., 2018; Mirman et al., 2018; Dvijotham et al., 2018) and those that formally verify adherence to specifications (Katz et al., 2017; Dvijotham et al., 2018; Singh et al., 2018). These works are orthogonal - our notion of verification differs from that of the software analysis/verification community that is concerned with showing models/pieces of code meet specifications for all inputs - as we aim to *learn* proof-verification protocols to solve a given task.

**AI Safety:** Being able to verify the correctness of the outputs produced by powerful but potentially adversarial or brittle learning systems is a top objective of the AI Safety community. Irving et al. (2018) propose a two player debate game where two agents communicate (space-constrained) statements to convince the judge of the correctness of opposing statements. Our approach is different in that it uses only a single debater (prover). We also differ significantly in our focus: while Irving et al. (2018) believed the presence of adversarial debaters would itself lend robustness, we are more interested in using the system to discover proof protocols. We believe the latter philosophy captures the spirit of the adversarial debate game from the rationalist community (Barnes & Christiano, 2020), which is intended to determine which forms of (human) argumentation lead most reliably to true conclusions. Che et al. (2021) propose Deep Verifier Networks, which uses a conditional variational autoencoder to detect "unreliable inputs or predictions." Unlike our approach, this approach hard-codes the structure of the verification algorithm, and considers out-of-distribution detection applications.

## 5 SIMULATIONS

### 5.1 PRACTICAL DESIGN DECISIONS

We elaborate on some design decisions we made for running prover-verifier training in practice.

**Practical Constraints on the Verifier:** If the verifier agent is instantiated using neural networks, then one can constrain the verifier by varying the architecture and/or reducing the size and depth of the network. While picking a suitable verifier constraint on toy problems might be tricky, we argue that this no longer remains an issue if the decision problem is inherently difficult to solve. For example, it is easy to find verifier architecture that cannot solve problems that have large search spaces (such as subset sum, SAT problems, or problems in NP in general), yet can check the proofs.

**Picking the Differentiable Game Optimizer:** We use standard gradient based optimizers (such as Adam (Kingma & Ba, 2014)) with alternating updates to find the equilibria of the prover-verifier game, since it was shown by Zhang et al. (2021) that simple alternating gradient descent-ascent is on par with more sophisticated algorithms for adversarial games. We suspect that some advances in the optimization of generative adversarial networks could help, and leave this as future work.

**Solving for Nash Equilibria:** Since computing the response gradients required to solve for Stackelberg equilibria is often computationally challenging, we opt to solve for Nash equilibria in our simulations. Since all Stackelberg equilibria of the proof-verification game are also Nash equilibria, it is a viable strategy to find a Nash equilibrium then check whether it also corresponds to a verifier-leading Stackelberg equilibrium. Also, the convergence properties of the simultaneous setup are favourable, as suggested by Theorem 4.

**Training Heuristics:** We list heuristics that improve speed and stability of training in Appendix D.

### 5.2 EVALUATING SUCCESS

The aim of the PVG is to produce a proof system which is both sound and complete. If this is the case, then we can expect the verifier to achieve high accuracy. However, the verifier's accuracy

---

[2]Probably Approximately Correct (Valiant, 1984)

Table 2: (**Precision and Recall on Binary Erasure Channel Task:**) We froze the verifier networks obtained via. Prover-Verifier Training and collaborative training and evaluated their precision and recall against both an optimized prover and directly optimized messages. The PVG-trained verifier achieved perfect specificity, whereas the collaborative-trained verifier achieved exactly 0 specificity.

| | **Frozen Verifier Recall** | | **Frozen Verifier Precision** | |
|---|---|---|---|---|
| | **PVG** | **Collaborative** | **PVG** | **Collaborative** |
| **Against Optimized Prover** | 1.0 | 1.0 | 1.0 | 0.0 |
| **Against Optimized Messages** | 1.0 | 1.0 | 1.0 | 0.0 |

during training does not constitute a good yardstick for evaluating success, since a high accuracy for the verifier could instead reflect problematic game dynamics. For example, high accuracy can be explained by the verifier detecting the prover's "tell" (spurious features in prover messages that correlate with the labels) rather than checking the proof. Also, we would like the proof protocol to generalize upwards to provers that are more capable than the one seen at training time.

Therefore, we propose to stress-test the robustness of the verifier by freezing it and giving a lot more resources to the prover. We did this in two different ways: First, we would freeze the verifier and continue to train the prover. Second, we would initialize messages to the prover's output and directly optimize the messages (to maximize the verifier's acceptance logit) using powerful optimizers, such as L-BFGS. Both of these techniques remove the verifier's ability to recognize the prover's tell, and also provide a degree of confidence in the robustness of the proof system. (It remains possible that there exist incorrect proofs which the verifier would accept but which are difficult to find.) We report the precision and recall of the frozen verifier against the two methods augmenting the prover.

The stress-tests outlined above can be seen as a way of checking if the verifier represents a verifier-leading Stackelberg equilibrium. Optimizing the prover corresponds to the setup where the problem instance is presented after agents pick their strategies. Optimizing the prover messages links to the setup where the instance is known to the prover (see Table 1).

We emphasize that our proposed evaluation methodology is essential to measuring progress on this problem: early on in our investigations, we were fooled into thinking the networks had learned a correct protocol because the verifier achieved high accuracy during training. However, we later determined that this high accuracy simply reflected the verifier learning the prover's "tell" faster than the prover learned to hide it. Since prior work aiming at similar goals to ours (Irving et al., 2018; Perez et al., 2019) reported only final accuracy during training, we consider our frozen verifier evaluation methodology an important contribution.

### 5.3 RESULTS

**Binary Erasure Channel Experiments:** We used the binary erasure channel (BEC) task described in Section 3.4 Definition 5 as a test bed to check whether our theoretical results on the desirability of equilibria are matched in practice. We parametrized the prover as a small fully connected network that takes one-hot-encoded input bits $[1, 0]$ or $[0, 1]$ as inputs and outputs logits over the $K = 16$ communication tokens. The verifier was instantiated as a small network that takes in the one-hot encoded communication tokens as inputs and outputs logits that are used to compute the probability the verifier assigns to the labels 0 and 1. We sampled the input bits uniformly, which means the labels were balanced. We used the Adam optimizer (Kingma & Ba, 2014) with its PyTorch defaults (Paszke et al., 2019), batch size of 2000 and learning rate of 0.0003. We updated the verifier 5 times for every prover update. Details about training setup are described in Appendix E.1.

To assess completeness and soundness, we computed the frozen verifier precision and recall against 1) a fully optimized prover and 2) fully L-BFGS optimized message vectors. The details for how we optimized the prover and message vectors are given in the Appendix E.1. We compared these results with those obtained using prover and verifier agents trained fully collaboratively (i.e. prover's labels always matches with the verifiers). The results, shown in Table 2, reaffirm our hypothesis that the PVG-trained frozen verifier achieved perfect precision and recall - both against the optimized prover and optimized messages. The collaboratively trained verifier had zero precision, hence is not robust.

In Figure 2, we visualize how the outputs (i.e. logits) of the prover and verifier networks for each different input evolve over time, accompanied by the accuracy of the verifier. We highlight - with bolded blue and orange curves - how the prover and verifier treat the special "0" and "1" tokens. The

Table 3: (**Precision and Recall on Find-The-Plus Task:**) We froze the verifier networks obtained via Prover-Verifier Training and collaborative training and evaluated their specificity and recall against both an optimized prover and directly optimized prover messages. The PVG-trained verifier achieved near-perfect specificity against the optimized prover and optimized messages. The specificity of the baseline was zero in all cases.

|  | **Frozen Verifier Recall** | | **Frozen Verifier Precision** | |
| --- | --- | --- | --- | --- |
|  | **PVG** | **Collaborative** | **PVG** | **Collaborative** |
| **Against Optimized Prover** | 1.0 | 1.0 | 1.0 | 0.0 |
| **Against Optimized Messages** | 1.0 | 1.0 | 0.99 | 0.0 |

verifier quickly picks up on the fact that receiving a "0" is indicative of the label being "0". Shortly thereafter (around iteration 400) the verifier learns the desired verification protocol.

**FindThePlus Experiments:** We tested the efficacy of prover-verifier games on a visual search task we call "FindThePlus". As in Figure 1, the dataset consists of binary $10 \times 10$ blue-red images that contain a single $3 \times 3$ plus pattern in them. Labels are determined by the color of the plus: images with a red plus are assigned to class 1 and images with a blue plus are assigned to class 0. The verifier is constrained to only process a portion of the input image, so it depends on the prover to advise it "where to look" (advice which it is free to ignore). We parametrized the prover as a convolutional net followed by a fully connected layer. We parametrized the verifier with a Spatial Transformer Network-like (Jaderberg et al., 2015) architecture with multiple heads.

The training details are provided in Appendix E.2. The frozen verifier precision and recall corresponding to Prover-Verifier training, as well as collaborative training are shown in Table 3. We optimized the prover and the messages using the same setup we used in the Binary Erasure Channel experiments. As before, the verifier trained under the prover-verifier game dynamics achieves perfect precision against the optimized prover, while the verifier trained under the standard, collaborative learning dynamics achieves exactly 0 precision. The frozen verifier precision was 0.99 if we directly optimized the proof vectors. We visualized where the Prover-Verifier trained verifier looks using the proof vectors from the prover. An example is displayed in Figure 1 (more in Appendix E.2). These vi-

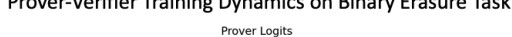
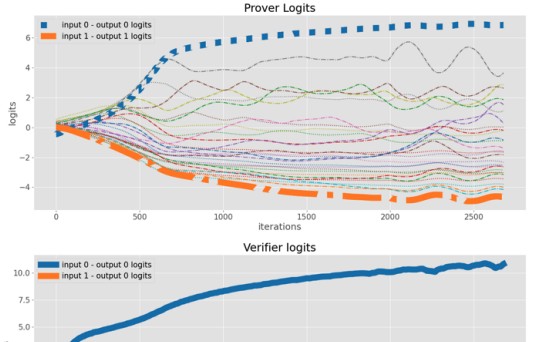
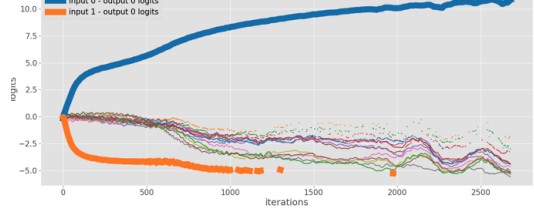
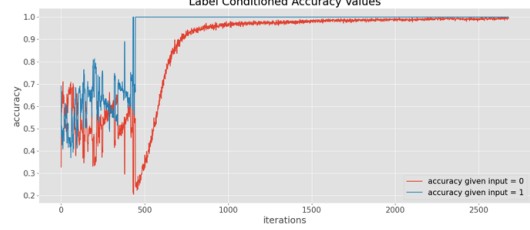

Figure 2: **BEC training dynamics:** The evolution of the logits, per input-output pair, of the prover (top) and verifier (middle), accompanied by the accuracy of the verifier (bottom). The verifier quickly picks the correct verification protocol, which drives the whole system into the desired equilibrium.

sualizations indicate that not only does the prover send the coordinate of the blue plus whenever it exists, it also sends the coordinates of maximally convincing patches if there's not a blue plus in the image.

## 6 CONCLUSION

We proposed Prover-Verifier Games (PVG), a game-theoretic framework to encourage neural networks to solve decision problems in a verifiable manner. We provided a thorough analysis of the problem space, and eliminated game formulations that have undesirable equilibria. Using a rigorous evaluation framework, we demonstrated - both theoretically and empirically - that there exist PVG formulations that can learn robust proof-verification protocols in practice.

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

## A    WHEN LOG-LOSS GRADIENTS VANISH

The gradients of the log-loss (i.e. the loss function used in Equations equation 3 and equation 4) vanish if and only if the agents fulfill their incentives (i.e. classify their labels correctly). Let $\boldsymbol{\ell} = v(x, p(x)) \in \mathbb{R}^d$ be the logits that the verifier network outputs for input $x$ and prover message $p(x)$. Let $S : \mathbb{R}^d \mapsto \mathbb{R}^d$ be the softmax operation with temperature 1: $[S(\boldsymbol{\ell})]_i = \frac{e^{\ell_i}}{\sum_{j=0}^d e^{\ell_j}}$. We take $S(\boldsymbol{\ell})_i$ to be the probability the verifier assigns for the correct label to be $i$.

Let $y$ be the correct label. The verifier loss and its gradient with respect to the logits can be given by:

$$\mathcal{J}_v = -\log \frac{e^{\ell_y}}{\sum_j e^{\ell_j}} \tag{7}$$

$$\frac{\partial \mathcal{J}_v}{\partial \boldsymbol{\ell}} = -(1 - \frac{e^{\ell_y}}{\sum_j e^{\ell_j}}) \tag{8}$$

If the verifier assigns a high probability to class $y$, then $\frac{e^{\ell_y}}{\sum_j e^{\ell_j}}$ will approach 1 and the gradients will vanish. Otherwise, the gradient won't vanish. The derivation is identical for prover's loss.

## B    PROOFS OF EQUILIBRIUM THEOREMS

**Theorem 1.** *Consider all prover-leading sequential PVG formulations. Having reached a (prover leading Stackelberg) equilibrium is neither necessary nor sufficient for having found a complete and sound proof-verification protocol.*

*Proof.* It suffices to prove that the prover-leading Stackelberg equilibrium is neither necessary nor sufficient for having found a complete and sound proof-verification protocol on the following decision problem:

**Definition 6** (Simplified Binary Erasure Channel Problem)**.** *Let $\mathcal{X} = \mathcal{Y} = \tilde{\mathcal{Y}} = \{0, 1\}$ and $\mathcal{M} = \{0, 1, 2\}$. Let the input distribution $\mathbb{P}_X$ be the uniform distribution over $\mathcal{X}$, and let the label generating function be defined as $y = f(x)$. That is, the label is identical to the input. The prover is arguing for label $y' = 1$.*

*The prover can choose between two policies $p_1$ and $p_2$:*

$$p_1 = \begin{cases} 0 & x = 0 \\ 1 & x = 1 \end{cases} \qquad p_2 = \begin{cases} 2 & x = 0 \\ 2 & x = 1 \end{cases}$$

*The verifier can pick between three policies $v_1$, $v_2$ and $v_3$ defined as follows:*

$$v_1 = \begin{cases} 0 & m = 0 \\ 1 & m = 1 \\ 0 & m = 2 \end{cases} \qquad v_2 = \begin{cases} 0 & m = 0 \\ 0 & m = 1 \\ 0 & m = 2 \end{cases} \qquad v_3 = \begin{cases} 1 & m = 0 \\ 1 & m = 1 \\ 1 & m = 2 \end{cases}$$

*Note that the mappings available to the verifier ignore the input. Also note that the prover-verifier pair $(p_1, v_1)$ satisfy Equations 5 and 6.*

*Lack of Necessity:* To show lack of necessity, it suffices to show that the complete and sound prover-verifier pair $(p_1, v_1)$ don't constitute a prover-leading Stackelberg equilibrium. If the prover picks $p_1$, the verifier can attain exactly 0 loss by assigning a weight of 1 to $v_1$ and a weight of 0 to $v_2$ and $v_3$. In this scenario, the prover's loss will be infinity:

$$\mathcal{J}_p = \underbrace{-\frac{1}{2} \log v_2(x = 0, p(x) = 0)}_{+\infty} + \underbrace{-\frac{1}{2} \log v_2(x = 1, p(x) = 1)}_{0} = +\infty$$

On the other hand, if the prover picks policy $p_2$, then its messages have exactly 0 mutual information with the input. In this scenario, the best the verifier can do to minimize its loss function is to assign equal probability to both outputs 0 and 1. It can do this by assigning 0 weight to $v_1$ and weights 0.5 to $v_2$ and $v_3$. The prover's loss in this scenario is:

$$\mathcal{J}_p = \underbrace{-\frac{1}{2} \log 0.5}_{\text{when x=0}} + \underbrace{-\frac{1}{2} \log 0.5}_{\text{when x=1}} = 1$$

Since the prover can do better by picking $p_2$ over $p_1$, the pair $(p_1, v_1)$ doesn't constitute a prover-leading Stackelberg equilibrium.

*Lack of Sufficiency:* Consider the scenario discussed above where the prover picks $p_2$, and the verifier responds by assigning 0.5 weight to $v_2$ and $v_3$. This is a Stackelberg equilibrium despite not corresponding to a complete and sound decision rule. This is because 1) $p_1$ achieves the best prover loss given that the verifier □

**Theorem 2.** *Consider all verifier-leading sequential PVG formulations in which the problem instance is revealed after the verifier picks its strategy. Having reached a (verifier leading Stackelberg) equilibrium is both necessary and sufficient for having found a desirable proof-verification protocol.*

*Proof.* We first prove necessity, then sufficiency.

*Necessity:* By the problem definition, there exists a prover-verifier pair $(p_1, v_1)$ that satisfy Equations 5 and 6, therefore constitute a complete and sound proof-verification protocol. Consider the scenario where the prover's policy is $p_1$, and the verifier assigns a weight of 1 to policy $v_1$. These prover-verifier strategies constitute a (Stackelberg) equilibrium of the verifier-leading sequential game formulation. This is because (1) $p_1$ already achieves the smallest prover loss for the given verifier strategy (2) the verifier already achieves 0 loss and therefore has no incentive to change its strategy. To see (1), note that the verifier can never be fooled if the correct label is 0, so the prover cannot possibly lower its loss by changing to a different strategy. If the correct label is 1, then the prover achieves 0 loss by following $p_1$. Therefore, the prover has no incentive to change its strategy.

*Sufficiency:* To show that having reached a verifier-leading Stackelberg equilibrium implies a complete and sounds proof-verification protocol, we follow a "proof by contrapositive" argument and prove, instead, that if the verifier is not representing a complete and sound proof-verification protocol, it cannot possibly be a Stackelberg equilibrium.

Assume that the verifier assigns a non-zero weight to a verification policy $v^{**}$ that is neither complete and/nor sound. If it's not complete, then for $v^{**}$ there exists a label-1 input $x$ such that there doesn't exists a prover policy that can convince the verifier that the correct label is 1. This will cause the verifier's loss to be nonzero. If it's not sound, then there exists a label-0 input for which there exists a prover policy that can fool the verifier to answering 1. This will also cause the verifier's loss to be nonzer. On the other hand, if the verifier assigns 0 weight to all non-complete nor non-sound verification modules, then it's loss will be exactly 0. This implies that the verifier can do strictly better by switching to a complete and sound verification protocol. □

**Theorem 3.** *Consider the simultaneous PVG formulation where the problem instance is revealed after the agents pick their strategies. Having converged to a (Nash) equilibrium is necessary for having found a complete and sound proof-verification protocol.*

*Proof.* Proving necessity for the simultaneous setup is identical to proving necessity of the verifier-leading Stackelberg setup.

The complete and sound prover-verifier pair $(p_1, v_1)$ (i.e. the verifier strategy of assigning a weight of 1 to $v_1$), whose existence is guaranteed by the problem definition, form a Nash equilibrium.

If we fix the verifier strategy at $v_1$, then the prover has no incentive to change it's strategy. In the collaborative case (when the input belongs to class 1, the prover already achieves 0 loss. In the adversarial case, the verifier is never fooled (by the soundness assumption) so the prover cannot improve its loss by switching to a different strategy. Therefore, if we fix the verifier strategy, the prover wouldn't move.

If we fix the prover strategy at $p_1$, then the verifier already achieves 0 loss by following $v_1$. Therefore it doesn't have an incentive to move.

Therefore, the $(p_1, v_1)$ pair form a Nash equilibrium. $\qquad\square$

**Corollary 1.** *All verifier-leading Stackelberg equilibria are Nash equilibria of PVG.*

*Proof.* Having reached a verifier-leading Stackelberg equilibria is sufficient for having found a complete and sound proof-verification protocol (by Theorem 2). A sound and complete proof-verification protocol already constitute a Nash equilibrium 3. Therefore, having reached a verifier-leading Stackelbeg equilibrium implies having reached a Nash equilibrium. $\qquad\square$

## C  PROOFS FOR EQUILIBRIUM ANALYSIS

**Theorem 4.** *Starting from any initialization with $p_i^0 > 0, p_i^1 > 0$ for all $i \in \mathcal{M}$, running gradient descent on both prover and verifier with reasonable learning rates, the prover would converge to only sending message 1 when $x = 1$ while the verifier would predict 1 only if it receives message 1 from the prover. To put it differently, $p_1^1 = 1$ when converge while $q_1 = 1 - \epsilon$ and $q_0 = q_2 = ... = q_K = \epsilon$ with $\epsilon$ a small constant depending on the strength of entropy regularization.*

*Proof.* First, we can assume WLOG that $K = 2$ because all these tokens from 2 to $K$ are "exactly" same for both prover and verifier. So if we do not concern about computation time, we can group tokens $2 - K$ as a single token. In this case, the only parameters for the prover is just $p_0^0$ and $p_1^1$. For notational convenience, we let $b \triangleq p_0^0$ and $a \triangleq p_1^1$. Further, we assume the learning rates $\eta_\mathcal{P} \ll \eta_\mathcal{V}$. With two time-scale updates, one can show that the verifier would converge first. Given nondegenerate initialization and entropy regularization on $q$ with coefficient $\lambda > 0$ (this is basically label smoothing), we have

$$q_1 = \frac{1 + \lambda}{1 + 2\lambda}, \quad q_0 = \frac{\lambda}{1 + 2\lambda}, \quad q_2 = \frac{1 - a + \lambda(2 - a - b)}{(2 - a - b)(1 + 2\lambda)}. \tag{9}$$

Recall that the objective of the prover is

$$\mathcal{L}(a, b) = -a \log q_1 - b \log q_0 - (2 - a - b) \log q_2 \tag{10}$$

Hence, we have the mean gradients of the prover

$$\frac{\partial \mathcal{L}}{\partial a} = \log q_2 - \log q_1 = \log \frac{1 - a + \lambda(2 - a - b)}{(1 + \lambda)(2 - a - b)} = -\log\left(1 + \frac{1 - b}{1 - a + \lambda(2 - a - b)}\right) \tag{11}$$

$$\frac{\partial \mathcal{L}}{\partial b} = \log q_2 - \log q_0 = \log \frac{1 - a + \lambda(2 - a - b)}{\lambda(2 - a - b)} = -\log\left(1 - \frac{1 - a}{1 - a + \lambda(2 - a - b)}\right) \tag{12}$$

As long as $a < 1$, $b$ would keep decrease as the gradient w.r.t $b$ in equation 12 is $> 0$. This further implies that the gradient w.r.t $a$ is $< 0$ and hence $a$ would keep increasing until $a = 1$.

Further, we show $a = 1, b < 1$ and $q$ taking the values in equation 9 are the Nash equilibria of the game. It is easy to see the $q$ value are optimal given fixed $a, b$, so it is impossible to improve unilaterally for the verifier. On the other hand, we know $\log q_1 > \log q_0 = \log q_2$ if $a = 1$ and $b < 1$, so the prover could not improve either. This completes the proof.

$\square$

**Theorem 5.** *Starting from any initialization with $p_i^0 > 0, p_i^1 > 0$ for all $i \in \mathcal{M}$, running gradient descent on both prover and verifier with any learning rates, prover would converge to only sending message $2$ to $K$ no matter what the input is.*

*Proof.* From the viewpoint of prover, it is essentially single-objective optimization problem with the loss function in the following form:

$$\mathcal{L}(a, b) = -a \log \frac{1 + \lambda}{1 + 2\lambda} - b \log \frac{\lambda}{1 + 2\lambda} - (2 - a - b) \log \frac{1 - a + \lambda(2 - a - b)}{(2 - a - b)(1 + 2\lambda)} \quad (13)$$

Surprisingly, the Stackelberg game formulation leads to bad joint strategy where the prover may send wrong messages with the input $1$. In particular, we have the gradient:

$$\frac{\partial \mathcal{L}}{\partial a} = -\log \left( 1 + \frac{1 - b}{1 - a + \lambda(2 - a - b)} \right) + \frac{1 - b}{1 - a + \lambda(2 - a - b)} \quad (14)$$

$$\frac{\partial \mathcal{L}}{\partial b} = -\log \left( 1 - \frac{1 - a}{1 - a + \lambda(2 - a - b)} \right) - \frac{1 - a}{1 - a + \lambda(2 - a - b)} \quad (15)$$

By the identity $\log(1 + x) < x$, we immediately know that $\frac{\partial \mathcal{L}}{\partial a} > 0$ and $\frac{\partial \mathcal{L}}{\partial b} > 0$ if $a, b \neq 1$. Interestingly, this is exactly the opposite of the simultaneous game. This immediately implies that $a, b$ will converge to $0$. This completes the proof. $\square$

## D  TRAINING HEURISTICS

We find that the following design decisions make learning more consistent and help speed up convergence, while not being essential for success:

- **Different Number of Gradient Steps per Agent:** We find that updating the verifier multiple times per a single prover update helps. It is especially helpful in preventing the verifier from ignoring the prover's messages early on in training. Note that proving Theorem 4 requires setting a larger learning rate to the verifier.

- **Adaptive Prover Step Size:** Adaptively increasing the prover-update frequency if the verifier's accuracy surpasses a pre-selected (i.e. 90 percent) value helps speed up the transition from "nearly robust" protocols to fully robust ones at the terminal phase of the training.

- **Prover Pretraining:** Pretraining the prover to solve the decision problem before initiating the prover-verifier game ensures that the first messages the verifier receives from the prover will likely contain features that correlate with the labels and therefore decreases the likelihood that the verifier will ignore the prover's messages. We do this by appending two extra heads to the prover that are trained to do solve the task at hand, as well as to autoencode the input.

## E  EXPERIMENT DETAILS

### E.1  BINARY ERASURE CHANNEL EXPERIMENTS

- **Task Setup and Data Distribution:** We picked there to be 16 communication tokens that the prover can communicate to the prover. We sampled the input bits uniformly, which mean the labels were balanced.

- **Prover Network:** We parametrized the prover as a two hidden layer fully connected LeakyReLU network with width 100 and LayerNorm (Ba et al., 2016) layers fully connected network. We used the Gumbel-softmax trick (with a constant temperature of 1) to sample (one hot encoded) communication tokens. We use the Straight-Through Gumbel Softmax Estimator (Maddison et al., 2017) to backpropagate through the discrete sampling step.

- **Verifier Network:** The verifier was instantiated as a one hidden layer, fully connected LeakyReLU neural network with width 100 and LayerNorm layers.

- **Optimization setup**: We used the Adam optimizer for updating both the prover and verifier with a learning rate of 0.0003 and batch size of 2000. We updated the verifier 5 times for each prover update. We observed that training the prover to solve the classification and autoencoding auxiliary tasks doesn't have an impact on training, likely due to the simplicity of this task. We ran optimization until the verifier learned a complete and sound proof-verification protocol (or, frozen verifier's precision and recall reached the value of 1)

- **Evaluation details:** To train the prover against the frozen verifier, we use the Adam optimizer with a learning rate of 0.0003 for 500 gradient steps, which we found was enough to reach near-convergence. To optimize the prover messages directly, we use the PyTorch implementation (Paszke et al., 2019) of the LBFGS optimizer (Nocedal & Wright, 2006) with a learning rate of 1, history size of 300 and "strong Wolfe" line search. We terminate the optimization when either the L1 norm of the change in iterates becomes less than $10e-9$, or when the L1 norm of gradients become less than $10e-5$, or when the number of optimizer steps surpass 300. We found that the the tolerance-based termination criteria are met almost always before the maximum number of optimizer steps is reached.

- **Hyperparameter Optimization and Tuning:** We tuned the following aspects of training 1) Optimizer (Adam vs. SGD), 2) Learning rate (1e-3, 3e-4, 1e-4), 3) Batch size (500, 2000), 4) Network width (32, 100) 5) verifier number of steps (1, 5, 10, 20, 50). We found that using Adam, the larger batch size of 2000 and a larger width of 100 for the networks were crucial. Setting the learning rate to be $3e-4$ and verifier number of steps to 5 stabilized training and made it more consistent.

- **Training hardware:** Due to the low computational requirements of the BEC experiments, the experiments were run on an 2019 Modle MacBookPro with 8-Core Intel Core i9 processors.

### E.2 FINDTHEPLUS EXPERIMENTS:

- **Task Setup and Data Distribution:** We used $10 \times 10$ images with exactly one $3 \times 3$ plus patterns in them. The images were single-channel, and the pixel values were binary (i.e. $0$ or $1$). The plus patterns were either formed using $0$-pixels or $1$-pixels, which we refer to "red" or "blue" plusses in the original text for additional clarity. The used balanced labels during training.

- **Prover Network:** We parametrized the prover as a 3 hidden layer convolutional neural network with 40 hidden channels, kernel size of 3, padding of 1 and LeakyReLU (Maas et al., 2013) nonlinearities followed by Instance Normalization layers. We passed the output of this convolutional neural network through a mean-pooling operation follow by a linear projection layer that resulted in the message vectors. The message vectors were 32 dimensional.

  We also trained the prover network to both solve the decision problem itself (i.e. classification), as well as autoencode the inputs. We did this by tacking "auxiliary heads" that process the output of the convolutional backbone described above and produce their outputs. We used a linear classification auxiliary head. For the autoencoding auxiliary head, we used a Spatial Broadcast Decoder (Watters et al., 2019) with 3 convolutional layers with 32 channels, kernel size of 3 and LeakyReLU activations.

- **Verifier Network:** We parametrized the verifier with the Spatial Transformer Network-like (Jaderberg et al., 2015) architecture with 4 heads. This architecture linearly transforms the prover messages into $2 \times 3$ matrices that parametrize what kind of spatial transformation should be applied to the image. The transformed images are then fed into a final linear classifier that produces the verifier predictions. A schematic diagram of a single head of this architecture is displayed in Figure 3. We constrained the allowed spatial transformations to

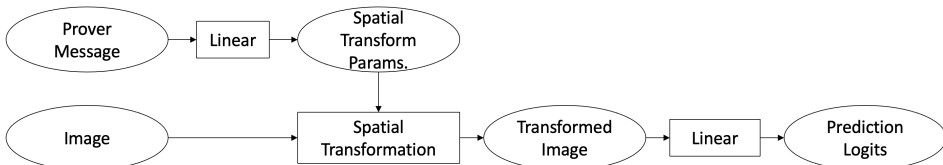

Figure 3: **FindThePlus Verifier Architecture:** A schematic diagram of a single head of the Spatial Transformer Network-like verifier architecture we used in out FindThePlus experiments.

only include shifts, and hard-coded the rest to zoom in by a factor of 2, and have no rotation and shear. These constraints only serve to improve the clarity plus visualizations and allow us to inspect training - training still succeeds if we remove these constraints. We used zero padding, if the verifier chose to shift its gaze to outside of the image.

- **Optimization Setup:** We used the Adam optimizer for updating both the prover and verifier with a learning rate of $0.0003$ and batch size of $2000$. We updated the verifier 1 times for each prover update. We observed that pretraining training the prover to solve the classification task speeds up training considerably, while the autoencoding head doesn't have a significant impact. We adaptively incremented the prover's number of steps if the verifier's accuracy kept above $75\%$ for 20 consequent game steps, and stopped incrementing when the prover's number of steps reached $15$. We ran pretraining using the same optimizer settings, for 100 gradient steps (enough to let the prover achieve around $80\%$ accuracy).

  We employed a bit of Label Smoothing (Müller et al., 2019) (with coefficient 0.05) to the verifier loss to regularize training. For pretraining the prover classification and autoencoding auxiliary heads, we used the cross entropy loss and (pixelwise) mean squared error respectively. We kept training the prover to classify and autoencode throughout training. We report our results at the point verifier achieved it's best frozen precision against directly optimized message vectors. Note that the verifier achieves perfect frozen precision against an optimized prover way before this point.

- **Evaluation Details:** We used exactly the same evaluation setup we used in the Binary Erasure Channel Experiments. We report the results on the training checkpoint with the highest overall frozen verifier accuracy against optimized proofs, as the performance of the verifier tends to fluctuate all throughout training. We also report the best result amongst training runs initialized with four random number generator seeds.

- **Hyperparameter Optimization and Tuning:** We used the setup that worked best for the Binary Erasure Taks and found that it also yielded good results for the FindThePlus task. We re-tuned the number of verifier steps and found that updating the verifier less frequently (i.e. once per prover step) doesn't degrade performance, while speeding up convergence. We also tuned the duration of prover pretraining. We found that pretraining the prover til convergence actually hurts performance. We link this to the Neural Collapse phenomenon elaborated by Papyan et al. (2020) and leave a more detailed examination to future works.

- **Training Hardware:** We ran training using a single NVIDIA P100 GPU with Intel Xeon Silver processors.

### E.3 ADDITIONAL PLUS VISUALIZATIONS

In Figure 4, we show more visualizations that display where the verifier looks at in the input images.

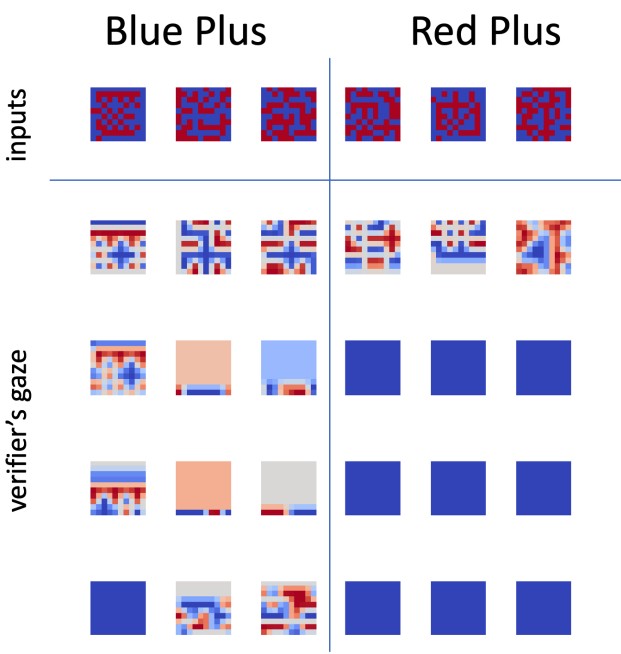

Figure 4: **Visualizing Verifier's Gaze:** We visualized where each of the verifier's 4 Spatial Transformer heads learn to look at. The top row displays the inputs. The left three inputs contain a blue plus in them (what the prover is defending). Each subsequent row corresponds to a different verifier head. We observe that the first head consistently contain blue plusses at consistent locations, indicating that the prover is communicating the coordinate of the plus to the verifier. If there's not a blue plus in the image, the prover sends the coordinate of a convincing image patch.

