# OpenReview forum: "Learning to Give Checkable Answers with Prover-Verifier Games"
_ICLR.cc/2022/Conference — ICLR 2022 Submitted_

### Official Review · Reviewer_cqwN · 2021-10-31

**Correctness:** 4
**Technical Novelty And Significance:** 3
**Empirical Novelty And Significance:** 2
**Recommendation:** 6
**Confidence:** 3

**Main Review:**

**Pros:**
- Coming up with a way to learn a network such that it is able to justify its answer is very important and useful in the real world
- It seems like the authors have done a good job analyzing the dynamics of the particular game in Table 1. I don't have a good sense of the significance of the theoretical contribution and would allow the other reviewers who are more familiar with the topic to comment on this point.

**Cons:**
- The toy examples seem too naive to demonstrate the effectiveness of the framework with respect to interpretability and robustness. Both of these directions are heavily researched, and the authors should be able to find other more realistic settings to convince us of the value of the framework.
- It is quite difficult for me to identify the practical usage of the framework when reading through the paper, perhaps the authors could have been more concrete with respect to the potential applications?

**Request for clarification:**

I am having some trouble understanding the practical usefulness of the proposed framework, and could really use some clarification from the authors. From my understanding, the main usages of the proposed framework are the **interpretability** and **robustness** of the network's decision. However, I don't think the experiments have demonstrated effectiveness in these two settings. Please correct me if I understand the purpose of the proposed framework incorrectly.

The authors try to demonstrate the interpretability of the prover's message by examing the gaze of the verifier, which depends on the prover's message. The author claims that the verifier's gaze is interpretable because it always focuses on the region of the image where a blue cross is present. However, the same behavior can be achieved quite easily with other techniques. For example, one can simply look at the saliency map of a naive classifier on the dataset and should be able to identify the cross-region quite easily or one can simply ablate each region of the image and study how the ablation changes the model's prediction.

With respect to robustness, the author demonstrated the robustness of the verifier's answer with respect to the prover's message, but I am having trouble understanding why being robust to the prover's message is important. For the most part, we care about robustness with respect to changes in the input distribution as machine learning systems are deployed in a different environment, but in what scenario would robustness to an internal prover's message be useful? Does robustness to the prover's message translate to robustness in settings that we care about?

-----updates after rebuttal------

After the author's clarification, I am able to understand the theoretical contribution more, and found it to be a valuable contribution. The empirical weakness of the paper remains. However, I have updated my scores to a 6.


**Summary Of The Paper:**

The authors propose a framework for training networks, such that justification of the answers can automatically emerge from it. Specifically, they propose a training framework where a prover's objective is to persuade the verifier network while the verifier network's objective is answering correctly based on both the original input and the prover's message. The authors analyze when the two-agents training network would converge and identify when the solution derived would become trivial. The authors also applied their framework in two experimental settings: the binary erasure task and the cross detection task, and show that the method would results in robust and interpretable decision rule.

**Summary Of The Review:**

The goal of coming up with a neural network that can justify its decision is an important task. The authors have done a thorough job of technically analyzing the proposed framework, but experiments could be strengthened by using more realistic settings.

---

> ### Author Response · Authors · 2021-11-15
> **Response to Reviewer cqwN**
>
> Thank you for your careful review. We appreciated that you found our approach well motivated and important, and our analysis of the PVG formulation comprehensive. We will now try to respond to your request for clarification, then address some of the cons you’ve listed.
>
> __Clarifications:__
> * __Practical usefulness of proposed framework:__ As we express in our general response, AI Safety is amongst our top motivations in exploring Proof-Verification Games. While we don’t provide instantiations of the Proof-Verification Game that would have direct practical applications in this paper, we believe we make a strong case that PVG will be useful in this direction:
>     * Our theorems about the desirability of equilibria show under which game setup and incentive structure the agents will learn to actually prove and verify. We believe this is a very nontrivial observation, and would be of interest to the AI Safety community.
>     * The coordinate-based proof-verification strategy in the FindThePlus task *emerges* from playing the PVG.
> * __Comment regarding interpretability of FindThePlus proofs:__ Our primary objective in analyzing the proof protocol in the FindThePlus task was 1) to show that the proof protocol learned is indeed complete and sound (i.e. the coordinate-based protocol can never go wrong) and 2) this protocol *emerges* as a result of the agents playing the Proof-Verification Game. Our primary objective with this analysis was actually *not* to demonstrate that the learned proofs are interpretable. The learned protocol, however, did end up being interpretable. This is not an accident, as the verifier architecture (Spatial Transformer Network), in informal terms, parametrizes a “gaze”, which makes it easy for humans to make sense of the proof. We think this workflow (designing verifier architectures that are aligned with human perception, and training it within the PVG framework) is an exciting research direction that the ICLR interpretability community would be interested in learning about.
> * __Comment regarding robustness:__ In cases where 1) the prover’s trustworthiness is uncertain 2) where we’d like to minimize false positive errors (such as in medical diagnosis), training a decision rule that’s complete and sound (and resistant to adversarial proof messages) would be of very high importance.
>
> * __Interpretability of proofs:__ A large number of problems can be decomposed into distinct search and verification steps. In interpreting the decision made by AI systems, the precise manner in which the search was computed is often irrelevant - however, the verification steps (complete with the certificate being verified) can tell us a lot.
> Monolithic networks combine these steps, making it difficult to pinpoint what decision procedure the network is following. Adding a *proof bottleneck* in the computational graph puts a particular structure on decision procedure and forces the prover to *communicate the certificate* directly. We have a toy example of this happening in the FindThePlus example - we’re guaranteed to directly find the coordinate information in the proof vector.
> How to extract interpretable information from real valued vectors is still a difficult problem, however knowing *which real valued vector* to focus on reduces the search space considerably.
> We also conjecture that learning to prove-verify will also bias the models towards learning more interpretable decision rules. We’re excited to explore this direction in the future.
>
> __Addressing critique:__
>
> * __Empirical results:__ The main motivations behind our experiments are
>     * To test (and consequently confirm) our theoretical results, and demonstrate that nothing unexpected happens when we instantiate the prover and verifier agents as neural networks
>     * To show (though to a limited extent) that unsupervised learning of proof protocols can happen, as it happens in the FindThePlus task
> Since the theoretical and empirical results show that our learning framework is sound, I think we present a strong signal to the ICLR community that the Prover-Verifier Game is interesting and promising.
> * __Practical use case:__ While we erred on the side of not overclaiming the potential use cases of our method in the paper, here are a couple practical applications where we hope and expect to see our system used for: 1) cases where false positives are very costly (like medical diagnosis), 2) Robust and interpretable vision systems 3) systems that involve interactions with very large networks that are very hard to trust/audit.

---

> > ### Author Response · Authors · 2021-11-19
> > **Follow-up**
> >
> > Thanks again for your time and careful review. As the interactive discussion period is coming to an end soon, we wanted to check if you have any remaining questions/requests for clarification. Specifically, please feel free to let us know if our response has clarified the motivation behind our work and highlighted why robustness against potentially adversarial provers is important.

---

> > ### Comment · Reviewer_cqwN · 2021-11-22
> > **Requesting further clarification**
> >
> > Thank you for the responses. Your answers are very helpful to me in understanding the contributions. However, I have a couple more questions that I would like to ask.
> >
> > You mentioned that the goal of analyzing FindThePlus proof is "to show that the proof protocol learned is indeed complete and sound (i.e. the coordinate-based protocol can never go wrong)", and you show that the proof protocol is sound by making sure that the verifier is robust to direct adversarial attack on the prover's message. However, even though adversarial attack is a very strong attack, it is possible that the strength of the attack is simply limited by the strength of the optimizer right? It is possible that with just the right optimizer or random initialization that we can indeed find a prover's message that fool's the verifier. It is also possible that you simply haven't tested the FindTheCross problem that is vulnerable to adversarial messaging. Due to the possibility (albeit small possibility), does that invalidate the claim that the empirically learned protocol is indeed sound?
> >
> > Even if the learned protocol in the specific game is shown to be indeed complete and sound, another learned protocol is not guaranteed to be complete and sound when applied to a more sophisticated problem, right? For example, when applying the same learning framework to medical analysis, it is possible that the learned protocol may be unsound especially with respect to new unseen medical images.

---

> > > ### Author Response · Authors · 2021-11-25
> > > **Further Clarification**
> > >
> > > Thank you for your questions.
> > >
> > > __Demonstrating completeness and soundness:__ To answer your first question, we first list different types of evidence we provide to show that PVG-learned protocols will be complete and sound. (all of these apply to the FindThePlus results)
> > > * __Theoretical results:__ Section 3.3 of the paper (PVG Formulations with Desirable Equilibria) outlines which PVG formulations will provably yield proof-verification behaviour, once an equilibrium is found. Section 3.4 expands this by showing how gradient based optimization can indeed find complete and sound proof-verification protocols.
> > > * __Empirical precision and recall values:__ On both FindThePlus and BinaryErasureChannel tasks, we demonstrate that the learned verifier is robust to adversarial proofs. We do this by both 1) optimizing the prover network and 2) directly optimizing proof messages, and demonstrating that the verifier is still not fooled. While this evaluation does not rule out the existence of proof vectors that will fool the verifier, we paid special attention to make sure that the “attacks” are as powerful as possible. Note that to construct the adversarial proof vectors, we used the highly effective, second order LBFG-S optimizer [1]. (To connect this measurement strategy to our theoretical results: This way of measuring completeness and soundness can be interpreted as checking whether the PVG has found a verifier-leading Stackelberg equilibria!)
> > > * __Confirming that the learned decision procedure is “right”:__ Visualizing the proof-verification protocol (as we did with FindThePlus experiments) gives us the chance to understand what “algorithm” the neural networks are approximating. We can then conclude if this algorithm constitutes a complete and sound decision rule. In the FindThePlus case, we can see that the learned protocol is based on communicating the coordinate of the plus. _If we implemented this algorithm using a standard programming language, like Python, we can prove without a doubt that the verifier will be complete and sound._ Note that there exist other protocols that yield perfect performance with a non-adversarial prover. For example, the prover can send “1” if the answer is True, and “0” elsewhere. This protocol achieves perfect performance with a non-adversarial prover, and exactly 0 precision when the prover acts adversarially.
> > >
> > > __On your second question:__ We apply the PVG on distinct _tasks_ that are defined by their input-output distribution. If we train Prover-Verifier networks on a medical diagnosis task that has a specific distribution, we can expect the learned protocol to be complete and sound if the game does converge. However, if the distribution of inputs changes during test time, this would mean the _task_ that the prover and verifier were trained on has changed, and we cannot expect the protocol to remain complete and sound. Please note that this limitation applies to virtually all learning-based methods.
> > >
> > > Please don't hesitate to follow-up if there remain further points that need clarification.
> > >
> > > [1] Nocedal, Jorge, and Stephen Wright. Numerical optimization. Springer Science & Business Media, 2006.

---

### Official Review · Reviewer_uyg3 · 2021-11-01

**Correctness:** 4
**Technical Novelty And Significance:** 2
**Empirical Novelty And Significance:** 2
**Recommendation:** 6
**Confidence:** 1

**Main Review:**

First of all, I have to admit that I am no expert in this area and hence I do not feel qualified to fully assess the merits of this paper. Also, for this reason, I find the paper really hard to follow. The presentation of the material does not offer a reader any help in this regard. The paper is not self-contained and a reader is supposed to go from one paper to another in hope of grasping what the authors are talking about.

Unfortunately, what is worse is that the paper delivers not a single example of a game that the authors are talking about. I do believe this makes the presentation suffer a lot (granted that I am from a different research area). The notation and the problem being studied aren't exemplified nor well-motivated either (as an example, what is PAC verification?).

Having said that, the paper seems to offer a rigorous theoretical study with a wealth of propositions, theorems and their proofs (there are several appendices attached). (In this sense, I am inclined to think that another venue might possibly fit this work better than ICLR.) Additionally, theoretical findings are confirmed by experimental analysis performed.

**Summary Of The Paper:**

Based on the ideas lying behind interactive proof systems (IPS), this paper proposes a prover-verifier games (PVG) framework, which aims at solving decision problems in a verifiable manner, where two learning agents interact with competing goals: a trusted verifier agent tries to get a correct answer to the problem while an untrusted (but more effective) prover tries to convince the verifier that its answer is correct, regardless of its actual correctness. The paper studies the framework under several (sequential and simultaneous) game scenarios.

**Summary Of The Review:**

To summarise, although I personally find the paper close to impossible to follow, I don't really feel in a position to judge it based solely on this fact simply because I am not an expert in the area, and it is unclear to me what presentation is deemed standard here and what level of preliminary descriptions is accepted as adequate.

---

> ### Author Response · Authors · 2021-11-15
> **Response to Reviewer uyg3**
>
> Thank you for your review and comments.
>
> We’ll try to act on your suggestion of augmenting exposition with the help of more concrete examples. While we consider which task best demonstrates the task we’re trying to address, here’s an example that might help put things in context:
>
> Let’s say our goal is to prove that an integer p is *not* prime. If p is indeed not prime, then we can easily demonstrate it by providing an integer n that divides p. This implies the following proof-verification strategy:
> * Preparation step: Prover and verifier receiver integer p.
> * Prover step: If p is not prime, the prover sends the verifier another integer n that divides p. Else, the prover sends an arbitrary integer.
> * Verifier step: Verifier concludes “Yes/accept” if n divides p (a cheap operation). Otherwise, if returns “No/reject”.
>
> This protocol was hand-engineered. Our goal is to come up with a framework to *learn* such proof-verification protocols. We establish theoretical results that outline under which conditions we can expect such learning to occur, and support these claims with simulations.
>
> (PAC stands for “Probably Approximately Correct” [1]. We’ll make sure to spell this abbreviation out in the revision!)
>
> We’re very happy to try and answer any question you have about our method, approach and proofs. Please feel free to let us know by commenting below.
>
> [1] Valiant, Leslie G. "A theory of the learnable." Communications of the ACM 27.11 (1984): 1134-1142.

---

> > ### Author Response · Authors · 2021-11-19
> > **Follow-up**
> >
> > Thanks again for taking the time to review our paper. Since the interactive discussion period is ending soon, please feel free to let us know if we can provide further clarifications regarding our motivation, framework and theoretical/empirical results.

---

### Official Review · Reviewer_wJaU · 2021-11-02

**Correctness:** 4
**Technical Novelty And Significance:** 2
**Empirical Novelty And Significance:** 2
**Recommendation:** 5
**Confidence:** 3

**Main Review:**

Strengths
- The paper does a good job of formalizing the various PVG settings and relates the effectiveness of the settings in realizing the goal of finding a desirable prover-verifier system. The analysis of the various equilibria in the context of completeness and soundness of the verifier provides good insight into the problem and might help future research.
- The empirical method of freezing the verifier to allow the prover more time to adjust to the verifier and hide its "tells" provides a more compelling case for the system's effectiveness.

Weakness
- Most of the theoretical analysis of achievable equilibria in the paper only applies to the case of BEC-based example prover-verifier systems. It is not clear if this would transfer well to more complicated settings. Although the theorems stated in the PVG formulation apply to a more general setting, the convergence guarantees for the games seem to need a lot more work.
- For the empirical evidence justifying the effectiveness of this strategy, I would like to see it applied to a more practical setting. The current approach seems to heavily depend on human intervention to limit the prover and verifier strategy space severely. I would like to see more theoretical work justifying the importance of expert intervention in the game design process or more experimental evidence in the absence of such interventions.

**Summary Of The Paper:**

The paper explores the idea of learning a game-theory-inspired prover-verifier system to augment neural networks with verifiable predictions.   In this paper, the authors set up a differentiable prover-verifier game and establish conditions on the formulation of the game to ensure the learned verifier satisfies the soundness and completeness constraints. The paper has the following results:
- All verifier leading Stackelberg equilibria in a verifier-leading sequential PVG formulation in which the problem instance is revealed after the verifier picks its strategy give a desirable proof-verification protocol.
- For the BEC-based prover-verifier system, the paper also guarantees that alternate gradient descent ascent converges to equilibrium under suitable learning rates verifier-leading PVG formulations.
- Empirical evidence from the BEC experiments and the FindThePlus experiments show the effectiveness of the learning scheme in a practical setting.

**Summary Of The Review:**

I feel the paper introduces a novel approach for learning prover-verifier systems. Still, I think the theoretical and experimental treatment of the topic in the current version of the paper is not rigorous enough at the moment. I feel the authors need to address the comments above.

---

> ### Author Response · Authors · 2021-11-15
> **Response to Reviewer wJaU**
>
> Thank you for your review and constructive critique.
>
> * __Strong formalization:__ We appreciate that you found our formalization of the PVG problem strong: In the early stages of this project, we spent a long time exploring directions that, due to our analysis, we now understand had been doomed to fail. We hope that our findings will help guide members of the ICLR community working on related problems or instantiations of Prover-Verifier games.
> * __Rigorous evaluation:__ We also thank you for picking out our evaluation strategy as an important contribution. Existing works in this area could benefit from similar evaluation strategies.
>
> We’d now like to address some points of critique:
>
> * __Convergence results__ (theoretical analysis of achievable equilibria is limited)
>     *  Our analysis of the properties of equilibria (Section 3.3) applies very broadly; it is only the convergence analysis (Section 3.4) that is limited to the BEC model. We believe such a restriction is necessary: proving global convergence results for games is very difficult without strong assumptions such as strong monotonicity. Finding Nash equilibria is a PPAD-hard problem [1] in the general setting. While our setting is not the fully general one, a global convergence result would imply, e.g., the ability to automatically discover primality certificates (since PRIMES is in NP). This seems too good to be true.
>     * Prover-Verifier Games is a very general framework that can be instantiated on many different kinds of learning problems. The binary erasure channel setup is simply a representation of the problem setup where 1) there does exist a complete and sound proof-verification protocol, and 2) the correct proof protocol can be accessed by the prover. Constructing a qualitatively different global convergence analysis would require making further assumptions about task structure, which we avoid in the confines of this paper.
>
> * __Importance of restricting verifier expressiveness:__ The restrictions imposed on the verifier simply serve to guarantee that it cannot solve the task on its own (i.e. absent the prover’s signal). While the examples in the paper required some unusual restrictions on the verifier, this was precisely due to the simplicity of the tasks. For more natural tasks (consider, e.g., theorem proving or Boolean satisfiability), ad-hoc restrictions would be unnecessary: it is easy to envision architectures sufficient to verify proofs, but much harder to build a prover network capable of solving the problem.
> * __Prover can be as expressive as needed:__ Our framework actually doesn’t require any restrictions on the prover strategy. Note that the prover agent in Interactive Proof Systems (a direct inspiration for our framework) is actually assumed to have unbounded capacity. (The constraint that exists in the Binary Erasure Channel task belongs to the communication channel between the prover and verifier, but not the prover. In the FindThePlus task, the prover communicates an unstructured real valued vector to the verifier, and there are no artificially imposed limitations on its expressive power)
>
> [1] Daskalakis, Constantinos, Paul W. Goldberg, and Christos H. Papadimitriou. "The complexity of computing a Nash equilibrium." SIAM Journal on Computing 39.1 (2009): 195-259.

---

> > ### Author Response · Authors · 2021-11-19
> > **Follow-up**
> >
> > Thanks again for your constructive critique and review. As we’re reaching the end of the interactive discussion period, we wanted to kindly ask whether our response addressed some of the concerns raised in your review. Please feel free to let us know if any issues remain and/or if there are any additional clarifications we can provide.

---

### Official Review · Reviewer_RcRe · 2021-11-03

**Correctness:** 3
**Technical Novelty And Significance:** 3
**Empirical Novelty And Significance:** 3
**Recommendation:** 6
**Confidence:** 3

**Main Review:**

Strengths:
- the new learning methodology is novel and built on top of solid theoretical foundations, namely, interactive proof systems and game theoretic optimizations;
- the effectiveness of different instantiations are backed up by both theoretical analyses and empirical studies;
- the writing is very easy to follow, even though there is quite an amount of theoretical stuff.

Weaknesses:
- chosen tasks in the evaluation seem fairly simple, so whether there is a great practical potential of the new methodology is not yet clear;
- the claim of learning in a "checkable" or "verifiable" manner could be somewhat misleading, as it seems _not_ really verifiable in the sense of classic software analysis and verification (i.e., a piece of code meets its formal specification for all possible executions). The naming might be standard in the theoretical computer science research community, which I am not quite familiar with;
- the motivation of learning robust protocols is not well-addressed. Basically, why does learning protocols matter in practice? The learned protocol and messages seem to be embedded in neural networks and real-valued vectors, so how could they be interpretable?


**Summary Of The Paper:**

This paper proposes a new learning methodology for training neural networks based on Prover-Verifier Games (PVGs), which are inspired by interactive proof systems (IPS). PVG consists of two learners, which work in both collaborative and adversarial manners, and the hope is that two learners together could achieve more reliable predictions. This paper studies eight possible game instantiations, depending on player order and when the problem instance is revealed, and the theoretical analysis shows that two instantiations are superior to others, which is also confirmed by an empirical evaluation. This paper also discovers that stress-testing of verifier's robustness is a more meaningful measurement of the learning success in contrast to prediction accuracy during training.

**Summary Of The Review:**

The idea proposed in this paper is novel, and its effectiveness is backed up by theoretical analysis and experimental evaluations. Although no strong practical results have been shown yet, the idea deserves to be published soon. I would like to recommend acceptance for this paper.

---

> ### Author Response · Authors · 2021-11-15
> **Response to Reviewer RcRe**
>
> Thank you for your encouraging review! We appreciate that you found our work novel, and based on solid foundations.
>
> We now try to address some of your critique:
>
> * __Claim of verifiability is too strong:__ Thanks for pointing this out. In the way it’s used in the paper, checkability has a precise definition in terms of the completeness and soundness of the learned protocol. Our frozen-verifier based evaluation methodology (which we consider an important contribution of our work) provides a highly stringent measure of how complete and sound a learned protocol is. That being said, we will modify the paper to make it clear that we use the term “verifiability” in a different sense than the way it’s used by the software analysis and verification community, as you’ve suggested.
> * __Simplicity of tasks:__ While our existing results don’t provide concrete instantiations of the PVG game that’d have direct practical applications,  we consider our existing results (empirical and theoretical) a strong signal that the PVG framework holds promise - something that we believe (and you’ve also alluded in your review!) the ICLR community would be interested in building on.
> * __Motivation:__ As mentioned in our general response, AI Safety is one of our top high-level motivations. While building interpretable systems is not our express motivation, we believe PVG also lends itself very nicely to this end goal. The fact that the proof protocol learned in the FindThePlus task is interpretable demonstrates that using verifier architectures that are aligned with human perception is a promising way of building interpretable proof protocols. One could also consider exploring different channel designs (i.e. natural language) to construct proofs, which we leave for future work.

---

> > ### Author Response · Authors · 2021-11-19
> > **Follow-up**
> >
> > Thanks again for your encouraging review! Since we’re reaching the end of the interactive discussion period, we wanted to kindly ask if you have any remaining questions/concerns that we can address. Please feel free to let us know.

---

### Author Response · Authors · 2021-11-15
**General Response**

We thank all reviewers for their thoughtful and constructive reviews.

We argue that Proof-Verification Games - a novel framework to learn proof-verification protocols - will interest the ICLR community that’s addressing the very important problem of training trustworthy models.

* __Novel framework and rigorous problem space analysis:__ Proof-Verification Game (PVG) is a novel learning framework to train trustworthy models. The equilibria of certain PVGs provably correspond to robust proof-verification protocols.
* __Merits of our experimental results vs. the difficulty of the problem:__ While judging the merits of the existing empirical results, we encourage the reviewers to consider what learning a fully robust proof-verification protocol entails. The verifier network, in our case, not only has to learn to identify verifiable features, but also be adversarially robust against perturbations on the prover’s messages. Hence, the fact that our PVG framework succeeded in learning completely robust proof-verification protocols, albeit on simplistic problems, is a promising signal - a signal which, we believe, the ICLR community would be interested in building on.
* __AI Safety as a Central Motivation:__ While a system whose internal decision procedure is robust and reliable would be useful on many generic tasks, we wanted to highlight AI Safety as a central high level motivation behind our work. It’s been observed that powerful AI systems such as large language models are prone to deception. [1] In such cases, we’re limited not by the power of the model but by the degree to which we can trust the results. While many different avenues to trustworthy AI deserve exploration, we think it’s especially promising to encourage the model to give its answers in a form that are easily checkable (either by a weaker, trusted model or by humans). For most questions we’d be interested in asking an AI, a verification protocol isn’t known in advance, and so it would need to be discovered with an approach such as ours. Our FindThePlus experiments shows that proof strategies could indeed emerge from playing the Proof-Verification game. In this experiment, the prover is not hard-coded to send the coordinate of the Pluses in the input images, but this is indeed the protocol that emerges.

__Summary:__ We propose a well-motivated and novel learning framework with promising applications, provide a rigorous analysis of its problem space, and establish tools to measure progress at it. Therefore, we argue that the ICLR community would be interested on building on our contributions.


[1]Evans, Owain, et al. "Truthful AI: Developing and governing AI that does not lie." arXiv preprint arXiv:2110.06674 (2021).

---

### Decision · Program_Chairs · 2022-01-20

**Decision:**

Reject

**Comment:**

The reviewers are split about this paper and did not come to a consensus: on one hand they agreed that the paper has valuable theoretical contributions and addresses an important problem in current ML literature, on the other hand they would have liked to see empirical results on a real-world problem setting. After going through the paper and the discussion I have decided to vote to reject for the following reason: I believe the reviewers' concerns about empirical results is not just a request for applying this to more datasets (which is easy to satisfy and I don't think is grounds for rejection), but is actually for a clearer connection for how this work would be used in the machine learning problems described in the introduction and related work sections. What would really help this paper is a real-world running example, in place of the blue plus example, in Figure 1 (I think the blue plus problem is still a useful experimental tool and should be evaluated, but it doesn't clarify the real-world use-cases of this work. This led the reviewers to look to the experimental section for clarification on this, but this wasn't clarified there either. The authors' response to these concerns was an out-of-scope argument: the goal of this paper is to derive/test theoretical results, and there are a number of possible use cases we could apply this to. The authors argue that the current work sends 'a strong signal to the ICLR community that the Prover-Verifier Game is interesting and promising'. I'm sorry but I disagree here: the authors need to do more to convince the ICLR community that this is a framework that will solve outstanding problems in ML. This is solved if the authors (a) run their approach on a real-world dataset in a paper they cite in the related work, (b) they include baselines in this experiment, and (c) if they add this as a running example throughout with a figure that explains this real-world example. With these additions the paper will be a much stronger submission.